# Genetic Diversity and Fingerprinting of 231 Mango Germplasm Using Genome SSR Markers

**DOI:** 10.3390/ijms252413625

**Published:** 2024-12-19

**Authors:** Jinyuan Yan, Bin Zheng, Songbiao Wang, Wentian Xu, Minjie Qian, Xiaowei Ma, Hongxia Wu

**Affiliations:** 1National Key Laboratory for Tropical Crop Breeding, Key Laboratory of Tropical Fruit Biology, Ministry of Agriculture and Rural Affairs, Key Laboratory of Hainan Province for Postharvest Physiology and Technology of Tropical Horticulture Products, South Subtropical Crops Research Institute, Chinese Academy of Tropical Agricultural Sciences, Zhanjiang 524013, China; yanjinyuan@webmail.hzau.edu.cn (J.Y.); zhengbin@catas.cn (B.Z.); wsbcjy@163.com (S.W.); xwt05@126.com (W.X.); 2Sanya Research Institute, Chinese Academy of Tropical Agricultural Sciences, Sanya 572024, China; 3National Key Laboratory for Germplasm Innovation and Utilization for Fruit and Vegetable Horticultural Crops, College of Horticulture & Forestry Sciences, Huazhong Agricultural University, Wuhan 430070, China; 4Sanya Nanfan Research Institute of Hainan University, Sanya 572025, China; minjie.qian@hainanu.edu.cn

**Keywords:** mango (*Mangifera indica* L.), genetic diversity, finger-printing, genomic SSR

## Abstract

Mango (*Mangifera indica* L.) (2n = 40) is an important perennial fruit tree in tropical and subtropical regions. The lack of information on genetic diversity at the molecular level hinders efforts in mango genetic improvement and molecular marker-assisted breeding. In this study, a genome-wide screening was conducted to develop simple sequence repeat (SSR) markers using the Alphonso reference genome. A total of 187 SSR primer pairs were designed based on SSR loci with consisting of tri- to hexa-nucleotide motifs, and 34 highly polymorphic primer pairs were selected to analyze the diversity of 231 germplasm resources. These primers amplified 219 alleles (*Na*) across 231 accessions, averaging of 6.441 alleles for per marker. The polymorphic information content (PIC) values ranged from 0.509 to 0.757 with a mean of 0.620. Genetic diversity varied among populations, with Southeast Asia showing the highest diversity, and Australia the lowest. Population structure analysis, divided the accessions into two groups, Group I (India) and Group II (Southeast Asia), containing 104 and 127 accessions, respectively, consistent with results from phylogenetic analysis and principal component analysis (PCA). Sixteen SSR primer pairs capable of distinguishing all tested accessions, were selected as core primers for constructing fingerprints of 229 mango accessions. These findings offer valuable resources for enhancing the utilization of mango germplasm in breeding programs.

## 1. Introduction

Mango (*Mangifera indica* L.) (2n = 40) is an important tropical perennial fruit, especially in tropical and subtropical regions [1], and is propagated by both sexual and vegetative propagation. It is hailed as the “King of Fruits”, celebrated for its rich aroma, vibrant skin, delicious taste, and high nutritional value [2]. The mango originated in the India–Myanmar region and spread to the Middle East, east Africa and South America around 300–400 AD [3]. During the 15th and 16th centuries, Portuguese and Spanish traders introduced mangoes to additional tropical and subtropical areas [4]. Today, mango is cultivated in over 100 countries, with two primary cultivar types: India and Southeast Asia, these types are distinguished by various of morphological characteristics [5,6,7]. Indian mango varieties generally ripen to orange or red, featuring round fruits and single embryo seeds. In contrast, Southeast Asian varieties often turn yellow or remain green, and have a distinctive “nose” or “beak”, along with polyembryonic seeds with multiple embryos [5,6]. China is one of major mango producing countries, with a total area of 395,200 hectares and an output of 4.32 million metric tons in 2022, according to the statistical data from Ministry of Agriculture and Rural Affairs. China preserves over 1000 mango germplasm resources, ranking second globally. This abundance presents challenges for identifying of mango varieties and analyzing their genetic characteristics [8]. Traditional morphological identification methods are unreliable due to their scarcity and susceptibility to environmental influences [9].

Molecular markers based on DNA polymorphisms, such as restriction fragment length polymorphisms (RFLPs), random amplified polymorphisms (RAPDs), single-nucleotide polymorphisms (SNPs), and simple sequence repeats (SSRs), are not affected by the growth period of plants. This makes them more suitable for identifying varieties than morphological markers [10,11]. SSRs and SNPs are two common molecular markers used in variety identification, genetic analysis, and fingerprinting, each offering distinct advantages [12,13]. SSRs have many advantages over other marker systems, including high reproducibility, polymorphic genetic information content, co-dominant nature, abundance, and good distribution throughout the genome, making them a valuable and effective tool [14]. SNPs are ideal for large-scale genome analysis and high-resolution research, and have high density, strong representation and good genetic stability throughout the genome, while SNPs are less polymorphic than SSRs markers due to their biallelic nature. In mango research, SSRs and expression sequence tag-derived SSR (EST-SSR) markers have been used utilized for genetic mapping [15], parent–child relationship allocation and validation [16], genetic diversity analysis [17,18], and fingerprint database construction [19,20,21]. Although SSR markers have proven effective in mango studies for decades, early SSR primers were mainly chosen based on polyacrylamide gel electrophoresis (PAGE), which resulted in limited data universality and comparability. SSR analysis of germplasm resources using capillary electrophoresis offers advantages such as high efficiency, stability, and sensitivity, which help minimize systematic and human errors [22]. Despite the abundance of mango SSR markers reported, most are based on PAGE detection, and there is still a significant need for suitable SSR markers for fingerprint database construction. Traditional methods of developing SSR markers, which involve constructing genomic libraries, are labor-intensive, time-consuming, and costly [23]. However, advancements in next-generation sequencing technologies have made SSR mining and marker development faster and more cost-effective [14]. Currently, mango SSR markers are mainly developed from transcriptome sequencing technology [24,25,26], but the relatively low polymorphism obtained based on transcriptome sequences hinders the method’s application of the method [27]. An alternative and efficient approach is to develop SSRs from publicly available genome sequences, thanks to the release of DNA sequence data from public databases [28]. The “Alphonso” mango genome (392.9 Mb) was released in 2020, enabling comprehensive analysis of mango genome-wide SSR loci characteristics and distribution patterns [29]. The objective of this study is to develop a set of mango SSR markers, evaluate the genetic diversity of accessions from different regions, and construct a fingerprint database for these accessions. The results will provide valuable insights for enhanced utilization of these accessions in mango breeding.

## 2. Results

### 2.1. SSR Locus Identification and the Frequency and Distribution of SSRs

A total of 109,177 SSR loci were identified using the whole mango genome sequences of “Alphonso”, with an average distance of 3.6 kb between loci. The frequency of SSRs repetition types decreases as the repeat unit increases. Mononucleotide repeats (49.94%) and dinucleotide repeats (27.32%) were the most prevalent, while trinucleotide repeats (10.37%), tetranucleotide repeats (8.26%), pentanucleotide repeats (3.07%), and hexanucleotide repeats (1.04%) were less common (Appendix A).

Among the SSR markers, six types of dinucleotide repeat motifs were identified: AT/AT, AG/CT, AC/GT, CG/CG, TC/GA, and TG/CA. The AT/AT motif was the most abundant, occurring 22,142 times and accounting for 74.23% of all dinucleotide repeats. Fourteen types of trinucleotide repeat units were identified, with AAT/ATT being the most frequent (6444 repeats, 56.89%). Other notable motifs included TTC/GAA (1223 repeats, 10.80%), AAG/CTT (1177 repeats, 10.39%), and ATC/GAT (473 repeats, 4.18%). For tetranucleotide repeats, 41 types were identified, with AAAT/ATTT being the most common (4207 repeats, 46.66%). Other frequent motifs included AATT/AATT (866 repeats, 9.41%) and ATAC/GTAT (707 repeats, 7.84%). Among pentanucleotide repeats, 97 types were identified, with AATAT/ATATT being the most common (547 repeats, 16.34%), followed by AAAAT/ATTTT (513 repeats, 15.33%) and AAAGG/CCTTT (320 repeats, 9.56%). For hexanucleotide repeats, 205 types were identified, with AAAAAT/ATTTTT being the most frequent (80 repeats, 7.04%), followed by AA-GATG/CATCTT (75 repeats, 6.60%) and ATCTTC/GAAGAT (49 repeats, 4.31%) (Figure 1).

### 2.2. SSR Primer Screening

A total of 12,009 primer pairs were designed using Primer 3 software for SSR loci with motifs ranging from tri- to hexa-nucleotides. To evaluate the efficiency and robustness of the primers and to select those with high polymorphism, 187 SSR primer pairs distributed over twenty chromosomes were designed and synthesized for three rounds of screening. In the first round, using the mixed DNA of twelve representative mango cultivars from different geographical origins as templates, 187 pairs were amplified and the allelic variation was detected by polyacrylamide gel electrophoresis detection methods, of which 157 pairs could be successfully amplified. Then, twelve representative mango varieties were used to screen the effectiveness of primers based on PAGE methods, and 81 SSR primer pairs were selected for primer re-screening. Figure 2 shows the PAGE results for primers G1134, G1342, G1382, and G1097 in the twelve mango varieties including Jinhuang, Tainong No. 1, Renong No. 1, Gedong, Keitt, Sri Lank No. 811, Nam Dok Mai Sitong, Dashehari, Zill, Haden, Guire No. 82, and Neelum. In the second round, the PCR products of 81 primer pairs in twelve varieties were observed whether the allele variation was the expected size, and the peak pattern was easy to identify based on capillary electrophoresis detection methods. Forty-five SSR primer pairs were selected for the second round of screening.

In the third round, 45 SSR primer pairs were selected for genetic diversity analysis based on capillary electrophoresis with 231 accessions. Ultimately, 34 pairs of primers were identified as the final core primers, on the basis of parameters such as polymorphism information content (PIC), allele count, peak reading clarity, and amplification stability. Figure 3 displays the capillary electrophoresis results of primer G573 across selected varieties (Figure 3).

### 2.3. Genetic Diversity

Thirty-four SSR primer pairs with high polymorphism were used to analyze genetic diversity analysis of 231 accessions (Table 1). The analysis revealed a total of 219 alleles were detected, with an average of 6.441 alleles per primer pair. The observed heterozygosity (*Ho*) was 0.707, while expected heterozygosity (*He*) was 0.672. The PIC values for the 34 SSR primer pairs ranged from 0.509 to 0.757, with a mean of 0.620. All selected primer pairs displayed PIC values exceeding 0.5, indicating that they are highly polymorphic. The average Shannon’s diversity index (*I*) for the population was 1.301. In summary, the analyzed accessions exhibited significant genetic diversity.

The genetic diversity of five populations—Australia, America, India, China, and Southeast Asia—was analyzed (Table 2). Southeast Asia exhibited the highest genetic diversity, with *I* (1.273), *He* (0.660), and PIC (0.680). This was followed by China, America, and India. Australia displayed the lowest genetic diversity, with *I* (0.874), *He* (0.529), and PIC (0.451). In all examined populations, *Ho* values exceeded *He* values, indicating an excess of heterozygotes.

### 2.4. Genetic Variation and Genetic Structure

The Analysis of molecular variance (AMOVA) revealed that within-population genetic variability accounted for 93.0% of the total variability, while among-population variability constituted 7% (Table 3). This pattern may result from a higher average gene flow between populations (*Nm* = 3.528).

The population structure of mango population was analyzed using a Bayesian clustering model. Following the method proposed by Evanno et al. [30], this study determined the optimal number of populations (K) for the 231 accessions based on the *ΔK* value. The estimated likelihood (LnP(D)) increased with K, and *ΔK* peaked at K = 2 (Figure 4). This indicates that the 231 accessions can be divided into two groups: Group I and Group II (Figure 5A). Group I comprised 104 accessions, including those from America (N = 38, 88.37%), India (N = 8, 72.73%), China (N = 29, 30.21%), Southeast Asia (N = 13, 22.41%), others (N = 9, 56.25%), and Australia (N = 7, 100%). Group II comprised 127 accessions, including those from America (N = 5, 11.63%), India (N = 3, 27.27%), China (N = 67, 69.79%), Southeast Asia (N = 45, 77.59%), and others (N = 7, 43.75%). Analysis of the Q values within each cluster revealed that 196 mango accessions (84.85%) had Q values greater than 0.6, with 108 accessions (46.75%) exceeding 0.9. This suggests that most of the tested germplasm has relatively uniform genetic backgrounds and a highly homogeneous relationship (Appendix A). Additionally, another peak was observed at K = 5. At this level, Group I was divided into subgroups 1 and 2, while Group II was split into subgroups 3, 4, and 5. However, individuals from different geographic regions did not exhibit a clear population structure and were distributed among the five subgroups.

The phylogenetic tree, constructed using the unweighted pair group method with arithmetic mean (UPGMA) method, is shown in Figure 5B. The results confirmed that the 231 accessions were divided into two major clusters, aligning with the population structure analysis. Most varieties from India, America, and Australia clustered together, while varieties from China and Southeast Asia formed a separate cluster.

Furthermore, a principal component analysis (PCA) was performed to validate the population structure results. The first and second principal components explained 9.76% and 7.25% of the molecular variance, respectively (Figure 5C). In summary, significant differences exist in the origins of various mango populations, which can be categorized into two groups: Indian and Southeast Asian varieties.

### 2.5. Construction of Fingerprint Database

To accurately identify and distinguish these germplasm resources, core primers were selected to construct their fingerprint database. The phylogenetic tree successfully differentiated 231 accessions. Among these, the pairs 37 and 67, 37 and 104, 94 and 182, 129 and 153, 136 and 153, and 143 and 162 showed differences at only one locus, suggesting potential similarities in the varieties. Field phenotype evaluations further indicated that 37 and 67, as well as 94 and 182, may represent synonyms, warranting the retention of one resource per pair. Following the principle of maximizing the number of distinguished germplasm while minimizing the number of primers, 16 SSR primer pairs were selected. These primers distinguished 229 accessions, which were used to construct their fingerprint profiles (see Table 4 and Appendix A). Additionally, a two-dimensional barcode fingerprint was created for the 229 test accessions (Figure 6 and Appendix A), incorporating information such as variety name, origin or source, and fingerprint code.

## 3. Discussion

Genome-wide identification of SSR loci and the development of SSR markers have been carried out in various species [28,31,32]. The mango genome was only published in 2020. Therefore, information on mango whole-genome SSR loci was relatively limited. In this study, 109,177 SSR loci were identified in the mango genome using Krait v1.3.3 software, with an average of one SSR locus per 3.6 kb, which is higher in quantity and density compared to results from transcriptome sequences [26,33]. While the mango genome contains an abundant number of SSR loci, significant differences were observed among nucleotide types. Mononucleotide SSRs were the most frequent, accounting for 49.94%, and the number of SSRs declined as the repeat unit length increased, consistent with transcriptome analysis [26]. A total of 365 types of repeat unit variations were identified across 1–6 nucleotides of the mango genome SSR sequences. Compared to other plants, these SSR motifs exhibited notable nucleotide preferences, with A/T being the dominant motif and G/C being relatively rare, aligning with studies on peony [34] and loquat [35]. The polymorphism of molecular markers is a crucial factor in evaluating SSR marker utility. However, stutter bands or peaks—resulting from enzyme slippage during amplification—complicate allele identification, particularly for SSR primers with dinucleotide motifs [36]. In contrast, markers with tri- to hexa-nucleotide motifs, especially those with fewer repeats, are less prone to stuttering issues [36,37]. To address this, 187 SSR primer pairs with tri- or higher nucleotide motifs were randomly selected and synthesized for validation of primer efficiency and polymorphism. After two rounds of screening, 34 primer pairs with stable amplification, high polymorphism, and good repeatability were chosen for genetic diversity analysis.

Genetic diversity, an important indicator of a species’ ability to adapt to environmental changes, is influenced by factors such as selection, genetic drift, migration, and breeding systems [38]. Studying the genetic diversity of mango germplasm is essential for variety identification and breeding new varieties. This study analyzed 231 accessions from populations across Australia, America, India, China, Southeast Asia, Cuba, et al. Genomic SSR molecular markers were employed to assess the genetic diversity of these mango accessions. A total of 219 bands were successfully amplified by 34 SSR primer pairs, with an average of 6.441 bands per primer pair. The PIC values for the SSR primer pairs ranged from 0.509 to 0.757, with an average of 0.620, higher than the value reported by Wang (0.490) [39], but lower than that reported by Tang (0.639) [20]. The average value of Ho, He, and I were 0.707, 0.672, and 1.301, respectively, indicating rich genetic diversity in the germplasm. Among the populations, the Southeast Asian population exhibited the highest genetic diversity, consistent with its role as a center of domestication [40]. Conversely, the Australian population displayed the lowest genetic diversity, aligning with previous analyses of Australian mango germplasm diversity. The genetic diversity of the “Kensington Pride” accessions is low with up to 38 accessions sharing all alleles [41]. In regions with limited genetic diversity, introducing germplasm with broader genetic backgrounds and creating new genetic variants through hybrid breeding are crucial strategies to enhance diversity for breeding and production [42]. The Indian population, another domestication center, did not exhibit the expected high genetic diversity, potentially due to selective germplasm. In contrast, the Chinese population demonstrated high genetic diversity, suggesting that local varieties could play a significant role in breeding programs. This study revealed that genetic variation among mango populations was relatively low, with 93% occurring within populations and only 7% between populations. The gene flow (*Nm*) value of 3.528 likely contributed to this limited genetic differentiation, as high gene flow tends to reduce variation between populations. Population structure analysis divided the 231 accessions into two groups corresponding to Indian and Southeast Asian varieties. Phylogenetic tree and PCA results were consistent, clustering American, Australian, and Indian varieties together, while most Chinese and Southeast Asian varieties formed a separate cluster. Overall, the 231 accessions were divided into two groups reflecting the two domestication centers of mango, India and Southeast Asia, consistent with findings from previous studies. Wilkinson et al. (2022) [43] analyzed 208 Australian mango accessions using 272 SNP markers and found that at K = 2, Southeast Asian accessions clustered independently, while other accessions formed a second group. Warschefsky and Wettberg (2019) [40] used SNP markers to analyze 106 mango cultivars from seven geographic regions and identified two gene pools representing Indian and Southeast Asian germplasm. Similarly, Sherman et al. (2015) [44] examined the Israeli mango germplasm collection, identifying two groups: one comprising Southeast Asian and Indian accessions and another comprising Floridian and Israeli cultivars. Multiple studies confirm the existence of two domestication centers for mango: India and Southeast Asia. Kuhn et al. [45] genotyped 1915 mango accessions from the United States, Thailand, and Australia using 272 SNP markers. Their work estimated genetic diversity, relatedness, and used a simple method to identify self-pollinated individuals and infer likely paternal candidates. However, these studies did not include Chinese cultivars. Recently, Ma et al. (2024) [46] and Liang et al. (2024) [47] independently analyzed the genetic diversity of Chinese mango accessions. Their findings revealed that Chinese accessions can be divided into two gene pools: Indian and Southeast Asian types. This study showed that most Chinese varieties cluster with Southeast Asian accessions, while a smaller subset clusters with American accessions. The predominant clustering with Southeast Asian varieties likely reflects geographical proximity and frequent gene flow. The clustering with American accessions may result from the influence of high-quality commercial varieties developed in the United States since the 19th century, as many Chinese cultivars trace their origins to these varieties. These findings suggest that Chinese mango germplasm has been domesticated from Southeast Asian and Indian lineages, with a stronger influence from Southeast Asia.

DNA fingerprinting, based on molecular markers, is a powerful tool for variety identification due to its speed, accuracy, high specificity, and environmental stability [48]. In this study, 16 SSR primer pairs were used to construct mango fingerprints. Based on allele size variations, a DNA fingerprint database was created for 229 mango accessions using a combination of letters and numerals. A two-dimensional barcode was also developed, incorporating information such as variety name, origin, and fingerprint code. This dual approach effectively resolves issues of synonymy and homology at the molecular level. On average, each primer pair distinguished 14.3 accessions, significantly improving the identification rate compared to previous studies [20,21]. This study developed efficient molecular markers and established digital and barcode fingerprint databases, creating a “digital identity card” for mango germplasm. By integrating phenotypic data, this platform supports rapid variety screening, accurate parent selection, and digital germplasm management. These resources are expected to aid the protection and precise identification of new mango cultivars. The SSR fingerprint database enhances variety identification efficiency and accuracy, providing a strong foundation for future studies on germplasm identification, protection of new varieties, and mango breeding.

## 4. Materials and Methods

### 4.1. Plant Materials and DNA Extraction

A total of 231 samples were used in this study, including 43 from America, 7 from Australia, 96 from China, 11 from India, 58 from Southeast Asia, and 16 from other countries. All materials were provided by the South Subtropical Crops Research Institute, Chinese Academy of Tropical Agricultural Sciences. Detailed origins are available in Appendix A.

Genomic DNA was extracted from young leaves using a modified cetyltrimethylammonium bromide (CTAB) method [49]. DNA concentration and purity were evaluated using a NanoDrop 2000 v3.3.1 ultramicro UV spectrophotometer (NanoDrop, Wilmington, DE, USA), and quality was confirmed by 1% agarose gel electrophoresis. The extracted DNA samples were diluted to a final concentration of 30 ng/µL and stored at −20 °C.

### 4.2. SSR Locus Identification and Primer Screening

Genomic sequences of the “Alphonso” mango, totaling 392.9 Mb [29], were analyzed for chromosome-specific perfect, compound, and imperfect SSR repeats using Krait’s ultra-fast SSR search module [50]. The recognition criteria for repeat units were as follows: mono-(12), di-(7), tri-(5), tetra-(4), penta-(4), and hexanucleotide repeats (4). SSR primers were designed with Primer 3 software under the following parameters: primer sizes ranged from 18 to 27 bp, annealing temperatures were set at 58–61 °C, GC content was maintained between 40% and 60%, and final product lengths ranged from 100 to 220 bp. A total of 187 SSR primer pairs, consisting of tri- to hexanucleotides, were randomly selected for validation and polymorphism testing. Initially, SSR primers were screened using 12 mango varieties from distinct geographical origins, followed by a secondary screening involving 231 mango resources. All primers were synthesized by Sangon Biotech Co., Ltd. (Shanghai, China) (Appendix A).

### 4.3. PCR Amplification and SSR Genotyping

PCR amplification was conducted in a 20 µL reaction volume containing 2 µL of DNA template (30 ng/µL), 10 µL of 2× Taq PCR Mix, 0.5 µL each of forward and reverse primers (10 µmol/L), and 7 µL of ultrapure water. The PCR procedure involved a pre-denaturation step at 94 °C for 5 min, followed by 35 cycles of denaturation at 94 °C for 30 s, annealing at 58 °C for 30 s, and extension at 72 °C for 30 s, concluding with a final extension at 72 °C for 8 min. All PCR reactions were carried out using a Bioer Life Pro Thermal Cycler (Bioer Technology, Hangzhou, China). The PCR products were detected with 8% non-denaturing polyacrylamide gels and capillary electrophoresis. Fluorescently labeled amplified products (ROX, HEX, or FAM) were multiplexed and separated on a 96-capillary automated DNA sequencer (ABI3130 DNA Analyzer, Applied Biosystems, Foster City, CA, USA). The PCR products were denatured, and DNA fragment sizes were assessed using the ABI3130 Capillary-based electrophoresis DNA sequencer. Fragment lengths were analyzed with GeneMapper v4.0 software. The resulting data were further analyzed using Peak Scanner v1.0 software (Applied Biosystems, Waltham, MA, USA) to determine the exact fragment sizes of the PCR products.

### 4.4. Data Analysis

The observed number of alleles (*Na*), effective number of alleles (*Ne*), *I*, *Ho*, *He*, genetic differentiation index (Fst), *Nm*, and AMOVA were performed using GenAlEx version 6.501 software [51]. The PIC, Nei’s genetic distance among mango individuals, and the construction of phylogenetic trees using the UPGMA were conducted with PowerMarker version 3.25 software [52]. Population structure analysis and the estimation of the optimal number of populations were completed using Structure version 2.3.4 software [30]. The burn-in period and the number of MCMC repetitions after burn-in were both set to 100,000, with K values ranging from 1 to 10. This process was repeated ten times for each K. The results were then uploaded to the Structure Harvester website (https://github.com/dentearl/structureHarvester/blob/master/harvesterCore.py) to determine the most suitable K value based on the *ΔK* method. The UPGMA evolutionary tree was enhanced using the ggtree v3.9.1 package. PCA and visualizations were performed using R version 4.3.3.

### 4.5. Construction of Fingerprints Database

To distinguish as many varieties as possible using the fewest primers, this study first selected partial primers capable of fully distinguishing the tested materials. These core primers were used to construct the fingerprint code for each tested material. Alleles from each primer pair were arranged in ascending order and assigned numerical values starting from 1. Values exceeding 9 were represented by letters starting with A, while the absence of a band was indicated by 0. Each germplasm at 16 SSR loci was assigned values based on a coding table (Table 5), resulting in a unique string known as the SSR fingerprint code. An online two-dimensional code generator (http://cli.im/) was used to create codes for each variety. The variety name, origin, and fingerprint code were inputted to generate a two-dimensional barcode for the tested mango.

## 5. Conclusions

In this study, 231 mango accessions were analyzed for genetic diversity using 34 SSR primer pairs developed from the Alphonso reference genome. The results revealed distant relationships and substantial genetic diversity among the accessions, which are important for the utilization of mango germplasm and breeding of new varieties in China. Additionally, fingerprint databases for 229 mango accessions were constructed using 16 core primers selected from the 34 SSR primer pairs, providing a technical foundation for future research on mango variety identification systems in China.

## Figures and Tables

**Figure 1 ijms-25-13625-f001:**
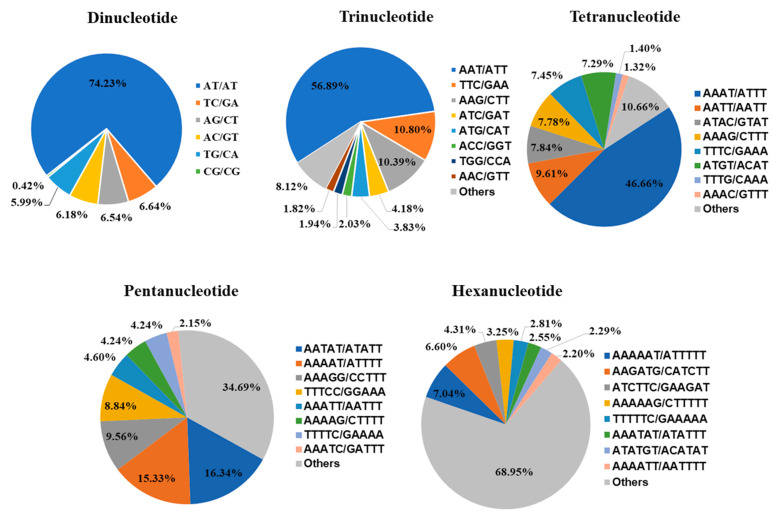
Proportions of different types of SSR units in the mango genome. Each pie chart represents the SSR loci with different repeat units in the genome.

**Figure 2 ijms-25-13625-f002:**
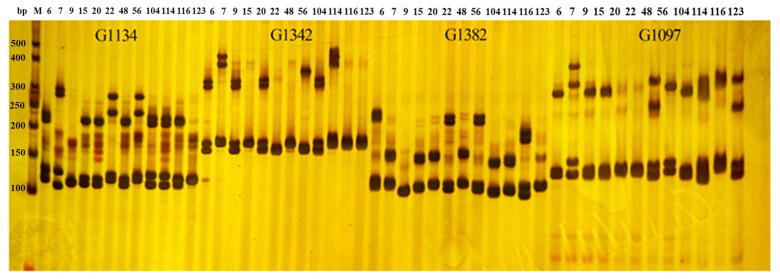
Polymorphism of 12 mango varieties by primer G1134, G1342, G1382, and G1097 based on PAGE. 6~123 represent the mango varieties Jinhuang, Tainong No. 1, Renong No. 1, GeDong, Keitt, Sri Lank No. 811, Nam Dok Mai Sitong, Dashehari, Zill, Haden, Guire No. 82, and Neelum, respectively.

**Figure 3 ijms-25-13625-f003:**
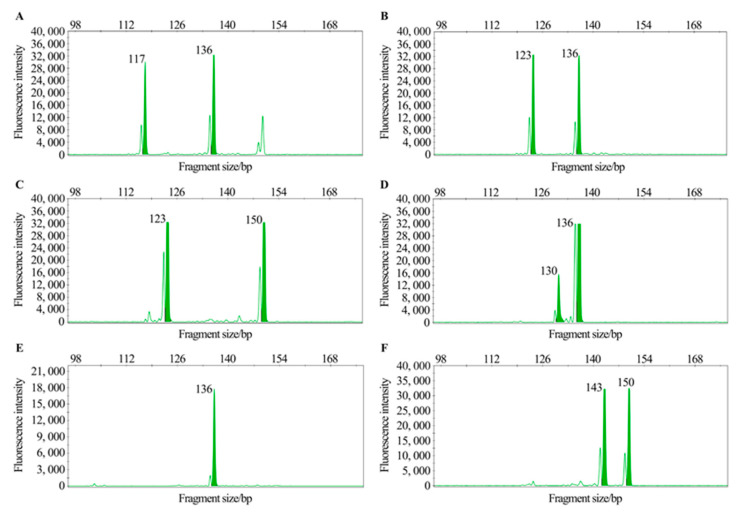
Different genotype of six mango varieties by primer G573 based on capillary electrophoresis. (**A**–**F**) represent the mango varieties Ono, Sri Lank No. 811, Glenn, Irwin, Renong No. 1, and Guifei, respectively.

**Figure 4 ijms-25-13625-f004:**
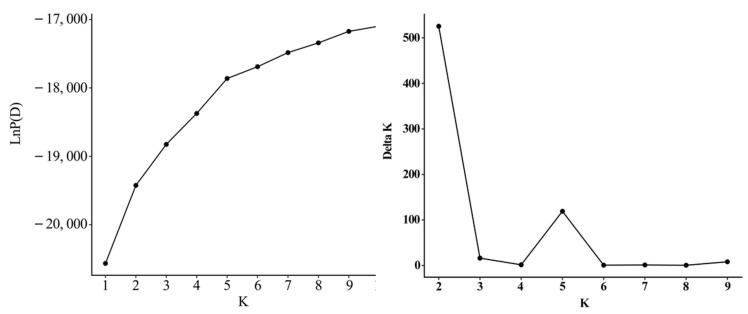
LnP(D) and *ΔK* evaluations of the 231 mango accessions.

**Figure 5 ijms-25-13625-f005:**
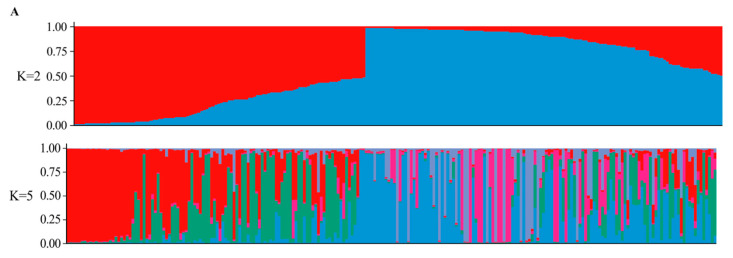
Genetic structure of 231 mango accessions. (**A**) Population structure analysis of the 231 mango accessions (K = 2 and K = 5). (**B**) UPGMA tree of 231 mango accessions based on Nei’s genetic distances. (**C**) The principal component analysis of the 231 mango accessions. All the results showed that the samples could be divided into two clusters: Red—Group I; Blue—Group II.

**Figure 6 ijms-25-13625-f006:**
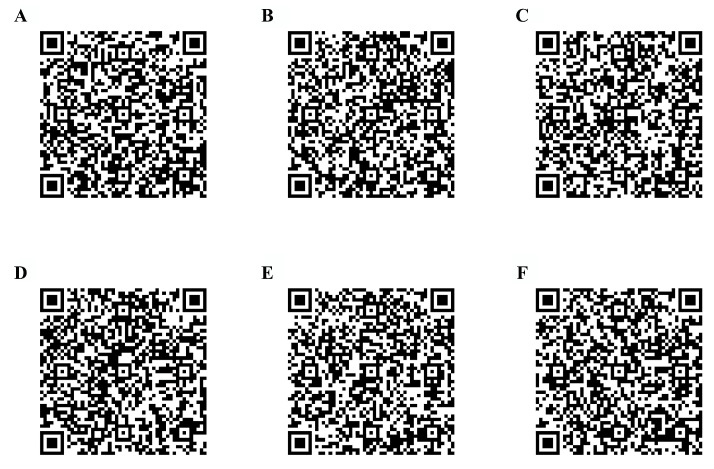
Fingerprinting of 2D barcode from part mango varieties. (**A**–**F**) represent the mango varieties Irwin, Apple, NamDocMai, Chishu, Arumanis B, and JinHuang, respectively.

**Table 1 ijms-25-13625-t001:** The values of genetic diversity statistics for 34 pairs of SSR markers.

Locus	*Na*	*Ne*	*I*	*Ho*	*He*	PIC
G33	8	2.926	1.326	0.654	0.658	0.6103
G111	10	2.508	1.380	0.623	0.601	0.5758
G216	7	2.748	1.251	0.736	0.636	0.5793
G304	6	2.379	1.041	0.632	0.580	0.5108
G386	4	3.009	1.195	0.749	0.668	0.6055
G440	4	2.669	1.039	0.649	0.625	0.5464
G573	8	2.639	1.314	0.606	0.621	0.5911
G584	7	2.794	1.236	0.628	0.642	0.5879
G589	5	2.749	1.154	0.641	0.636	0.5660
G676	7	3.560	1.476	0.784	0.719	0.6811
G744	10	3.860	1.556	0.844	0.741	0.7001
G824	8	2.336	1.071	0.710	0.572	0.5138
G833	8	2.517	1.133	0.571	0.603	0.5463
G886	6	2.831	1.311	0.619	0.647	0.6093
G1028	8	4.277	1.563	0.409	0.766	0.7284
G1077	9	3.424	1.510	0.688	0.708	0.6705
G1097	8	4.075	1.500	0.823	0.755	0.7114
G1134	6	3.351	1.335	0.740	0.702	0.6481
G1151	5	3.646	1.366	0.823	0.726	0.6766
G1212	4	3.294	1.262	0.684	0.696	0.6390
G1234	5	3.015	1.315	0.857	0.668	0.6280
G1268	3	2.380	0.967	0.658	0.580	0.5085
G1342	5	3.022	1.211	0.758	0.669	0.6063
G1352	5	3.264	1.254	0.489	0.694	0.6313
G1360	5	3.466	1.339	0.931	0.711	0.6565
G1382	6	4.736	1.632	0.874	0.789	0.7569
G1424	9	3.952	1.541	0.771	0.747	0.7073
G1450	7	2.554	1.205	0.615	0.609	0.5596
G1454	7	4.664	1.652	0.771	0.786	0.7534
G1527	6	3.018	1.233	0.749	0.669	0.6050
G1617	5	2.763	1.161	0.589	0.638	0.5734
G1666	6	3.189	1.280	0.797	0.686	0.6298
G1745	6	3.454	1.345	0.827	0.710	0.6588
G1783	6	2.472	1.077	0.727	0.596	0.5159
Mean	6.441	3.163	1.301	0.707	0.672	0.620

Note: *Na*: different number of alleles; *Ne*: effective number of alleles; *I*: Shannon’s information index; *Ho*: observed heterozygosity; *He*: expected heterozygosity; PIC: polymorphism information content.

**Table 2 ijms-25-13625-t002:** Mean values of the genetic diversity statistics for 34 SSR markers in five populations.

	*Na*	*Ne*	*I*	*Ho*	*He*	PIC	*Np*
America	5.353	2.947	1.225	0.726	0.644	0.589	11
Australia	2.971	2.226	0.874	0.773	0.529	0.451	1
China	5.441	3.027	1.243	0.710	0.655	0.601	3
India	4.206	2.923	1.168	0.660	0.628	0.573	0
Southeast Asia	5.676	3.120	1.277	0.687	0.661	0.610	11

Note: *Np*: private number of alleles.

**Table 3 ijms-25-13625-t003:** Analysis of molecular variance (AMOVA) of 231 mango accessions.

Source	df	SS	MS	Est. Var.	Genetic Variation Rate (%)
Among populations	5	353.667	70.733	1.485	7%
Within populations	225	4732.337	21.033	21.033	93%
Total	230	5086.004		22.517	100%

Note: df: degrees of freedom; SS: sum of squares; MS: mean square; Est. Var: estimated variance.

**Table 4 ijms-25-13625-t004:** Fingerprint code of some mango accessions.

No.	SSR Fingerprinting	No.	SSR Fingerprinting
1	35667755562411133444242411264555	6	45341555132445232355334615264635
2	25687715364414112311232533253655	7	34451155161424133444254558124656
3	24444555161225231315345615164445	8	25681118113355333434222518465646
4	25464558132315343435243588465666	9	36661613132212113444232215256656
5	25454655562644343344342336243635	10	35347715152424332315344518226635

Note: 1–10 represents the mango varieties Irwin, Apple, Nam Doc Mai, Chishu, Arumanis B, JinHuang, Tainong No. 1, kyo savoy, Renong No. 1, and Kent, respectively.

**Table 5 ijms-25-13625-t005:** Alleles encoded standard.

Primer	Code									
	1	2	3	4	5	6	7	8	9	A
G676	178	185	191	197	203	209	222			
G744	139	142	145	154	162	165	168	174	177	180
G1028	99	102	105	111	114	117	120	129		
G1077	119	122	125	128	131	134	137	140	145	
G1097	131	134	138	142	145	151	154	157		
G1134	106	112	119	126	133	139				
G1151	194	197	200	203	206					
G1212	135	138	141	149						
G1234	87	93	99	105	111					
G1352	130	133	137	144	147					
G1360	123	126	129	132	135					
G1382	104	107	110	113	116	119				
G1424	88	91	97	100	103	106	109	112	124	
G1454	119	122	125	128	131	134	137			
G1666	110	113	116	119	122	125				
G1745	118	121	124	127	130	133				

## Data Availability

Data are contained within the article and Appendix A.

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
