# Peer review of "Genetic Diversity and Fingerprinting of 231 Mango Germplasm Using Genome SSR Markers"

_ijms, 2024, doi:10.3390/ijms252413625_

Round 1
Reviewer 1 Report
Comments and Suggestions for Authors
The research investigates the genetic diversity of 231 mango germplasms using genome-wide SSR markers and constructs a fingerprint database for these accessions. The study aims to enhance the understanding of genetic diversity to support mango breeding programs and varietal identification.
The topic is highly relevant to the field of plant genetics and breeding, particularly for mango cultivation. It addresses a notable gap in developing SSR markers suitable for fingerprinting databases and genetic diversity analysis in mango. Although SSR markers have been utilized before, this study improves the methodology by leveraging genome-wide data from the Alphonso mango reference genome. This represents a novel and impactful contribution to the field.
The methodology is robust and leverages advanced techniques like capillary electrophoresis for SSR analysis.
The conclusions align well with the presented evidence and effectively address the main research questions.
The references are comprehensive and appropriate. They include seminal works, recent mango genetics advancements, and SSR marker development. However, additional citations to global studies on mango diversity could enhance the manuscript's contextualization.
Comments on Tables and Figures
Please double-check references to tables, figures and supplementary material in the text. There are inconsistencies (e.g. references to supplementary files S5 and S6 do not seem correct).
In addition: Figure 1 (SSR unit proportions) could benefit from clearer labeling or color-coding for better readability.
Including a high-resolution map showing the geographical origins of the accessions would enhance the context.
Additional Comments
The construction of a two-dimensional barcode system is innovative and practical for germplasm management. Further details on its potential integration into digital breeding platforms could add value.
The manuscript could benefit from a deeper discussion on the implications of the observed genetic diversity for mango breeding programs, especially in regions with limited diversity.
Author Response
Comments 1: Please double-check references to tables, figures and supplementary material in the text. There are inconsistencies (e.g. references to supplementary files S5 and S6 do not seem correct).
Response 1: Agree. We have checked references to tables, figures and supplementary material, see supplementary file which marked red in the text and new revision of supplementary file.
Comments 2: In addition: Figure 1 (SSR unit proportions) could benefit from clearer labeling or color-coding for better readability.
Response 2: Agree. We have revised the Figure 1 in the manuscript. The details see Figure 1 in the manuscript
Comments 3: Including a high-resolution map showing the geographical origins of the accessions would enhance the context.
Response 3: Thank you for pointing this out. We agree with this Comments.
A total of 231 accessions from different geographical origins were used in this study, including 43 from the USA, 7 from Australia, 96 from China, 11 from India, 58 from Southeast Asia, and 16 from other countries. A high-resolution map showing the geographical origin of the accessions would enhance the context, but citing or placing high-resolution map that includes China geographical in a paper or publication requires approval from a government agency, so we have not placed the high-resolution map in the manuscript.
Comments 4: The construction of a two-dimensional barcode system is innovative and practical for germplasm management. Further details on its potential integration into digital breeding platforms could add value.
Response 4: Thank you fro pointing this out. We agree with this Comments.
We have added the details of two-dimensional barcode system on its potential integration into digital breeding platforms in the manuscript.
This study developed efficient molecular markers and established digital and barcode fingerprint databases, creating a "digital identity card" for mango germplasm. By integrating phenotypic data, this platform supports rapid variety screening, accurate parent selection, and digital germplasm management. The details see 339-343 lines in the manuscript.
Comments 5: The manuscript could benefit from a deeper discussion on the implications of the observed genetic diversity for mango breeding programs, especially in regions with limited diversity.
Response 5: Agree. We have added the correlated contents in the manuscript.
The details see 291-293 lines in the manuscript.
Comments 6: The references are comprehensive and appropriate. They include seminal works, recent mango genetics advancements, and SSR marker development. However, additional citations to global studies on mango diversity could enhance the manuscript's contextualization.
Response 6: Agree. We have added the global studies on mango diversity in the manuscript. The details see lines 307-330 in the manuscript.
Reviewer 2 Report
Comments and Suggestions for Authors
Genetic diversity and fingerprinting of 231 mango germplasm using genome SSR markers
Authors Jinyuan Yan , Bin Zheng , Songbiao Wang , Wentian Xu , Minjie Qian , Xiaowei Ma * , and Hongxia Wu *
Introduction”
“Therefore, it is necessary to develop new SSR markers. It 76”
I disagree, SSR markers are still subject to challenges in building global databases. Most if not all would concur that SNP are the optimal marker system. I would have concluded ( and indeed have in prior publications on soybean and maize ): Therefore, it is necessary to develop SNP markers”
Nonetheless kudos to you all for all the hard work!
However, You must speak to several previous publications using SNPs:
Liang et al 2024 Front. Plant Sci., 02 July 2024
Volume 15 - 2024 | https://doi.org/10.3389/fpls.2024.1328126
Kuhn et al 2019 Estimation of genetic diversity and relatedness in a mango germplasm collection using SNP markers and a simplified visual analysis method Scientia Horticulturae Volume 252, 27 June 2019, Pages 156-168
Wilkinson et al 2022 The influence of genetic structure on phenotypic diversity in the Australian mango (Mangifera indica) gene pool Scientific Reports volume 12, Article number: 20614 (2022)
Sherman et al 2015 Mango (Mangifera indica L.) germplasm diversity based on single nucleotide polymorphisms derived from the transcriptome BMC Plant Biology volume 15, Article number: 277 (2015)
In particular How do the associations of mango genotypes shown by Liang et al , Kuhn et al , Wilkinson et al, Sherman et al ( who also used SSRs) compare with your findings?
Population structure and genetic diversity of mango (Mangifera indica L.) germplasm resources as revealed by single-nucleotide polymorphism markers
Genetic diversity
You need to explain why there are multiple bands per single primer pairs. Unless you have performed genetic analyses to determine nos of loci and which alleles belong to which locus then usage of the term “genetic” is more akin to “phenotypic” ALTHOUGH yes you are reflecting the products of genetic loci.
Discussion
carried out in many species with genomes [28,31,32]. Due to the lack of mango genomic 239
Correct the English!! Show me a species that does not have a genome!
Author Response
Comments 1: “Therefore, it is necessary to develop new SSR markers. 76”
I disagree, SSR markers are still subject to challenges in building global databases. Most if not all would concur that SNP are the optimal marker system. I would have concluded ( and indeed have in prior publications on soybean and maize ): Therefore, it is necessary to develop SNP markers”
Nonetheless kudos to you all for all the hard work!
Response 1: Thank you for pointing this out. We agree with this Comments.
We have revised it in the text: SSRs and SNPs are two common molecular markers used in variety identification, genetic analysis, and fingerprinting, each offering distinct advantages. SNPs are ideal for large-scale genome analysis and high-resolution research; however, they come with high development costs and complex data processing. In contrast, SSRs are more economical, exhibit greater polymorphism, and are easier to apply, making them a valuable effective tool.
The details see lines 61-66 in the manuscript.
Comments 2:
However, You must speak to several previous publications using SNPs:
Liang et al 2024 Front. Plant Sci., 02 July 2024,Volume 15 - 2024 | https://doi.org/10.3389/fpls.2024.1328126
Kuhn et al 2019 Estimation of genetic diversity and relatedness in a mango germplasm collection using SNP markers and a simplified visual analysis method Scientia Horticulturae Volume 252, 27 June 2019, Pages 156-168
Wilkinson et al 2022 The influence of genetic structure on phenotypic diversity in the Australian mango (Mangifera indica) gene pool Scientific Reports volume 12, Article number: 20614 (2022)
Sherman et al 2015 Mango (Mangifera indica L.) germplasm diversity based on single nucleotide polymorphisms derived from the transcriptome BMC Plant Biology volume 15, Article number: 277 (2015)
In particular How do the associations of mango genotypes shown by Liang et al , Kuhn et al , Wilkinson et al, Sherman et al ( who also used SSRs) compare with your findings? Population structure and genetic diversity of mango (Mangifera indica L.) germplasm resources as revealed by single-nucleotide polymorphism markers
Response 2: Thank you for pointing this out. We agree with this Comments.
We have cited relevant literature and compared and discussed it with them. For detailed information, please see the discussion section (lines 307-330).
Comments 3: Genetic diversity:You need to explain why there are multiple bands per single primer pairs. Unless you have performed genetic analyses to determine nos of loci and which alleles belong to which locus then usage of the term “genetic” is more akin to “phenotypic” ALTHOUGH yes you are reflecting the products of genetic loci.
Response 3: Agree.
Multiple bands per single primer pair due to low primer specificity or problems with PCR reaction parameters may result in some blurred or unspecific bands, and the true bands are the size of the target product when designing the primer.
Genetic diversity refers to the germplasm resources that have a certain level of genetic diversity in terms of SSR variation.
Comments 4: Discussion: carried out in many species with genomes [28,31,32]. Due to the lack of mango genomic 239. Correct the English!! Show me a species that does not have a genome!
Response 4: Agree. We have revised it: Genome-wide identification of SSR loci and the development of SSR markers have been carried out in various species. The mango genome was only published in 2020. Therefore, information on mango whole genome SSR loci was relatively limited.
Reviewer 3 Report
Comments and Suggestions for Authors
Yan et al.’s manuscript details the development of SSR markers to explore the genetic diversity and create a fingerprinting system for 230 mango accessions. This study also established a comprehensive mango fingerprinting database. The findings provide valuable insights into the genetic background of mango breeding materials, offering mango breeders essential tools for more informed breeding strategies and genetic resource management. Few minor adjustments are listed below:
Line 80. Please double check the citation, ‘[ ]’ is missing for the reference.
Line 330. Can you explain why this recognition criteria is used?
Line 328. What is 392.9Mb29?
Line 342. How much total amount of DNA templated were added to the reaction?
Line 212. PCA?
Author Response
Comments 1: Line 80. Please double check the citation, ‘[ ]’ is missing for the reference.
Response 1: Agree. We have Revised [29]. The details see line 87 in the manuscript.
Comments 2: Line 330. Can you explain why this recognition criteria is used?
Response 2: We used the criteria for screening genomic SSRs based on the SSR search criteria recommended by Krait software and with reference to the crops such as sugarcane, peach, tartary buchwheat and carya.
Comments 3: Line 328. What is 392.9Mb29?
Response 3: Agree. We have revised it in the manuscript: Genomic sequences of the ‘Alphonso’ mango, totaling 392.9 Mb [29]. (see lines 361).
Comments 4: Line 342. How much total amount of DNA templated were added to the reaction?
Response 4: Agree. We have revised the DNA template: 2 µL of DNA template (30 ng/µL) (see lines 376)
Comments 5: Line 212. PCA?
Response 5: Agree. We have changed PCoA to PCA, PCA is principal component analysis. (see lines 222).
Reviewer 4 Report
Comments and Suggestions for Authors
This is an interesting manuscript with a clear objective, well-defined methodology, and high practical relevance. I have only a few minor suggestions for improvement:
1. In Figure 2, for each SSR target fragment, such as G1134 with a target size of 127, there are fragments around the range of 200-250 that exhibit similar band intensities to the target fragment. The occurrence of such multiple bands should be clarified.
2. Regarding the names of mango cultivars, please review them thoroughly. For instance, "Sir Lank No.811" should be corrected to "Sri Lanka No.811."
3. For population genetics parameters, it is common practice to italicize terms like Na, Ne, I, Ho, He, etc.
4. In the STRUCTURE analysis, while the highest Delta K value was found to be 2, the second-highest Delta K value is 5. Therefore, the analysis results for a K value of 5 should also be presented in Figure 5 for a more complete interpretation.
5. Similarly, in the STRUCTURE analysis section, relevant terms should be italicized in accordance with the software’s guidelines.
I do not have specific comments on the English language; however, I would recommend that the authors have their manuscript polished by a native English speaker or a professional editing service to enhance its clarity and readability.
Author Response
Comments 1:In Figure 2, for each SSR target fragment, such as G1134 with a target size of 127, there are fragments around the range of 200-250 that exhibit similar band intensities to the target fragment. The occurrence of such multiple bands should be clarified.
Response 1: Agree. In figure 2, nonspecific bands appear in SSR primer amplification,the true bands are the bands within the target fragment range of the primer.
Comments 2: Regarding the names of mango cultivars, please review them thoroughly. For instance, "Sir Lank No.811" should be corrected to "Sri Lanka No.811."
Response 2: Agree. We have revised and correct the names of mango cultivars in the whole manuscript and supplementary file.
Comments 3: For population genetics parameters, it is common practice to italicize terms like Na, Ne, I, Ho, He, etc.
Response 3: Agree. We have changed the genetics parameters to italicize terms.
Comments 4: In the STRUCTURE analysis, while the highest Delta K value was found to be 2, the second-highest Delta K value is 5. Therefore, the analysis results for a K value of 5 should also be presented in Figure 5 for a more complete interpretation.
Response 4: Agree. Additionally, We have added the correlated contents in the manuscript. The details see lines 194-198 in the manuscript.
Additionally, another peak was observed at K=5. At this level, Group I was divided into subgroups 1 and 2, while Group II was split into subgroups 3, 4, and 5. However, individuals from different geographic regions did not exhibit a clear population structure and were distributed among the five subgroups.
Comments 5: Similarly, in the STRUCTURE analysis section, relevant terms should be italicized in accordance with the software’s guidelines.
Response 5: Agree. We have changed the relevant terms to italicize terms in STRUCTURE analysis section.
Comments 6: I do not have specific Commentss on the English language; however, I would recommend that the authors have their manuscript polished by a native English speaker or a professional editing service to enhance its clarity and readability.
Response 6: Agree. We have revised the manuscript and polished it by a professional editing service. please see the revised manuscript.
Round 2
Reviewer 2 Report
Comments and Suggestions for Authors
positively addressed reviewers comments
Author Response
Comments 1: “Therefore, it is necessary to develop new SSR markers. 76”
I disagree, SSR markers are still subject to challenges in building global databases. Most if not all would concur that SNP are the optimal marker system. I would have concluded ( and indeed have in prior publications on soybean and maize ): Therefore, it is necessary to develop SNP markers”
Nonetheless kudos to you all for all the hard work!
Response 1: Thank you for pointing this out. We agree with this Comments.
We have revised and added it to the introduction section of in the manuscript:
SSRs and SNPs are two common molecular markers used in variety identification, genetic analysis, and fingerprinting, each offering distinct advantages. SNPs are ideal for large-scale genome analysis and high-resolution research; however, they come with high development costs and complex data processing. In contrast, SSRs are more economical, exhibit greater polymorphism, and are easier to apply, making them a valuable effective tool.
Comments 2:
However, You must speak to several previous publications using SNPs:
Response 2 :
Thank you for pointing this out. We agree with these Comments.
We have cited, compared and discussed relevant literature. Detailed information can be found in the Discussion section.
Comments 2.1:
Liang et al 2024 Front. Plant Sci., 02 July 2024,Volume 15 - 2024 | https://doi.org/10.3389/fpls.2024.1328126
Response 2.1:
We cited the literature of liang et al.(2024)in the Discussion section of the manuscript.
For details, see:
Ma et al. (2024) [46] and Liang et al. (2024) [47] independently analyzed the genetic diversity of Chinese mango accessions. Their findings revealed that Chinese accessions can be divided into two gene pools: Indian and Southeast Asian types.
Comments 2.2:
Kuhn et al 2019 Estimation of genetic diversity and relatedness in a mango germplasm collection using SNP markers and a simplified visual analysis method Scientia Horticulturae Volume 252, 27 June 2019, Pages 156-168
Response 2.2:
We have cited the literature of Kuhn et al. (2019) in the Discussion section of the manuscript.
For details, see:
Kuhn et al. [45] genotyped 1915 mango accessions from the United States, Thailand, and Australia using 272 SNP markers. Their work estimated genetic diversity, relatedness, and used a simple method to identify self-pollinated individuals and infer likely paternal candidates.
Comments 2.3:
Wilkinson et al 2022 The influence of genetic structure on phenotypic diversity in the Australian mango (Mangifera indica) gene pool Scientific Reports volume 12, Article number: 20614 (2022)
Response 2.3:
We have cited the literature of Wilkinson et al. (2022) in the Discussion section of the manuscript.
For details, see:
Wilkinson et al. [43] analyzed 208 Australian mango accessions using 272 SNP markers and found that at K = 2, Southeast Asian accessions clustered independently, while other accessions formed a second group.
Comments 2.4:
Sherman et al 2015 Mango (Mangifera indica L.) germplasm diversity based on single nucleotide polymorphisms derived from the transcriptome BMC Plant Biology volume 15, Article number: 277 (2015)
Response 2.4:
We have cited the literature of Sherman et al.(2015) in the Discussion section of the manuscript.
For details, see:
Warschefsky and Wettberg [40] used SNP markers to analyze 106 mango cultivars from seven geographic regions and identified two gene pools representing Indian and Southeast Asian germplasm. Similarly, Sherman et al. [44] examined the Israeli mango germplasm collection, identifying two groups: one comprising Southeast Asian and Indian accessions and another comprising Floridian and Israeli cultivars.
Comments 2.5:
In particular How do the associations of mango genotypes shown by Liang et al , Kuhn et al , Wilkinson et al, Sherman et al ( who also used SSRs) compare with your findings? Population structure and genetic diversity of mango (Mangifera indica L.) germplasm resources as revealed by single-nucleotide polymorphism markers.
Response 2.5:
Kuhn et al. (2019) , Wilkinson et al. (2022), and Warschefsky and Wettberg (2019) used SNP markers to analyze population and genetic diversity and confirmed the existence of two domestication centers for mango: India and Southeast Asia. Sherman et al. (2015) [44] used SSR and SNP markers and identified 74 Israeli mango accessions into two groups: one comprising Southeast Asian and Indian accessions and another comprising Floridian and Israeli cultivars.However, but all these studies did not include Chinese germplasm. Ma et al. (2024) [46] and Liang et al. (2024) [47] independently analyzed the genetic diversity of Chinese mango accessions. Their findings revealed that Chinese accessions can be divided into two gene pools: Indian and Southeast Asian types. Our results showed that the 231 accessions could be divided into two groups: India and Southeast Asia, which was consistent with the results of previous studies, Further validating the results of Ma et al. (2024) [46] and Liang et al. (2024) [47].
Comments 3:
Genetic diversity:You need to explain why there are multiple bands per single primer pairs. Unless you have performed genetic analyses to determine nos of loci and which alleles belong to which locus then usage of the term “genetic” is more akin to “phenotypic” ALTHOUGH yes you are reflecting the products of genetic loci.
Response 3:
Agree.
Multiple bands per single primer pair due to low primer specificity or problems with PCR reaction parameters may result in some blurred or unspecific bands, and the true bands are the size of the target product when designing the primer.
Genetic diversity refers to the germplasm resources that have a certain level of genetic diversity in terms of SSR variation.
Comments 4:
Discussion: carried out in many species with genomes [28,31,32]. Due to the lack of mango genomic 239. Correct the English!! Show me a species that does not have a genome!
Response 4:
Agree.
We have revised it:
Genome-wide identification of SSR loci and the development of SSR markers have been carried out in various species. The mango genome was only published in 2020. Therefore, information on mango whole genome SSR loci was relatively limited.